# Comparison Study of PVD Coatings: TiN/AlTiN, TiN and TiAlSiN Used in Wood Machining

**DOI:** 10.3390/ma15207159

**Published:** 2022-10-14

**Authors:** Beata Kucharska, Paweł Czarniak, Krzysztof Kulikowski, Agnieszka Krawczyńska, Krzysztof Rożniatowski, Jerzy Kubacki, Karol Szymanowski, Peter Panjan, Jerzy Robert Sobiecki

**Affiliations:** 1Division of Surface Engineering, Warsaw University of Technology, Woloska 141, 02-507 Warszawa, Poland; 2Institute of Wood Sciences and Furniture, Warsaw University of Life Sciences, 159 Nowoursynowska Street, 02-776 Warsaw; 3Institute of Physics, University of Silesia, 75 Pulku Piechoty 1, 41-500 Chorzow, Poland; 4Jožef Stefan Institute, Jamova 39, SI-1000 Ljubljana, Slovenia

**Keywords:** PVD method, TiN/AlTiN, TiN, TiAlSiN coatings, tool durability tests, chipboard machining

## Abstract

In this paper, we analyze the possibilities of the protection of tools for wood machining with PVD (Physical Vapor Deposition) hard coatings. The nanolayered TiN/AlTiN coating, nanocomposite TiAlSiN coatings, and single layer TiN coating were analyzed in order to use them for protection of tools for wood machining. Both nanostructured coatings were deposited in an industrial magnetron sputtering system on the cutting blades made of sintered carbide WC-Co, while TiN single layer coating was deposited by evaporation using thermionic arc. In the case of TiN/AlTiN nanolayer coatings the thickness of the individual TiN and AlTiN layer was in the 5–10 nm range, depending on the substrate vertical position. The microstructure and chemical composition of coatings were studied by scanning electron microscopy (SEM) and energy-dispersive X-ray spectroscopy (EDS) method. Additionally, in the case of the TiN/AlTiN coating, which was characterized by the best durability characteristics, the transmission electron microscope (TEM) and X-ray photoelectron spectroscopy (XPS) methods were applied. The coatings adhesion to the substrate was analyzed by scratch test method combined with optical microscopy. Nano-hardness and durability tests were performed with uncoated and coated blades using chipboard. The best results durability characteristics were observed for TiN/AlTiN nanolayered coating. Performance tests of knives protected with TiN and TiAlSiN hard coatings did not show significantly better results compared to uncoated ones.

## 1. Introduction

Cemented carbides (CC) are used as tool materials due to their excellent properties. Different advanced techniques of materials modifications allow for the extension of the lifetime of CC blades used for wood machining. The most known methods for tool wear protection are PVD (physical vapor deposition) and CVD (chemical vapor deposition) techniques [1]. Comparing both methods PVD coatings can be harder, have finer grain size and smoother surface morphology. They can also be crack-free and have residual compressive stress. Therefore, the advantageous features of PVD coatings relative to CVD ones are in interrupted cut applications (e.g., milling, threading) and where sharp edges are needed (e.g., finishing-cut operations) [2].

CVD coatings proved better adhesion due to deeper atomic diffusion into the substrate at high deposition temperature [3,4]. Better adhesion is particularly important when sharp edges are required and in an interrupted cut application. However, CVD coatings have certain disadvantages in residual compressive stress and surface morphology [2]. The most commonly used coating materials in wood industry for cutting tools are based on nitrides and carbides: TiN [5,6], TiAlN [7,8,9], AlTiN [5,10,11], TiAlSiN [12] as well as CrAlN [5,13,14] and CrN [15,16,17,18]. Titanium nitride (TiN) exhibit high hardness, good adhesion for substrate, low friction coefficient, high wear resistance, and good oxidation resistance. Aihua et al. found that in the case of TiN coating deposited on the cemented carbide the main wear mechanism is oxidation and abrasive wear [5]. The mechanical and tribological properties of TiN coatings depends on the type of substrate on which it was deposited. It was proved that the wear mechanism of TiN coatings irrespectively of the substrate was abrasive wear [6].

Compared to TiN, the TiAlN coatings have greater chemical stability and higher hardness and oxidation resistance at high temperatures which results in better machining performance [9]. Tools with TiAlN coatings provide better crater wear resistance, increase wear resistance and support higher cutting temperatures [9]. Compared to other hard coatings, the results of Aihua et al. showed that TiN and TiAlN coatings presented lower friction coefficient and lower wear rate and that high Al content AlTiN and CrAlN coatings didn’t present better anti-wear properties in this test [5]. TiAlN possessed perfect tribological properties compared with the other coatings but the wear rate of TiN and TiAlN coatings had a lower value than AlTiN and CrAlN [5].

In the last decade, high-performance nanostructured PVD hard TiAlSiN coatings based on silicon addition have been proposed for the protection of cutting tools [7,12]. TiSiN and TiAlSiN coatings containing Si are often named “nanocomposite” because they consist of TiN nanocrystals embedded in an amorphous matrix of Si_3_N_4_ silicon nitride. Such nanocomposite coatings have high hardness, high oxidation resistance and high thermal stability [12].

Recently, the multilayers coatings such as TiN/CrN, TiAlN/CrN, TiN/TiAlN, TiN/AlTiN were studied due to their better properties compared to a single layer coating [18,19,20,21,22,23,24,25]. They exhibit higher hardness and wear and oxidation resistance. A large number of various coatings based on TiAlN and multilayer TiAlN/TiN were tested at Koszalin University of Technology by Warcholinski et al. [19]. The coatings were deposited seven bilayers, each of the thickness 400 nm. It was proved that the TiAlN coating shows a higher oxidation resistance than TiN. The previous authors work also dealt with multilayer coatings (TiN/TiAlN and TiAlN/a-C: N) [20]. In the article examining the effect of coating thickness on their properties, it was shown that the blades covered with 5 µm TiN/AlTiN coatings exhibited the best durability characteristic in the chipboard test. Based on these results, the authors made an attempt to evaluate the behavior of various coatings in the tests on the chipboard (TiN, TiAlSiN and TiN/TiAlN) with the extension of the tests with the XPS results.

There are many articles comparing the structure and properties of different types of coatings [5,20,22,23,26]. However, the available sources lack the durability results of cutting tools on a material that is extremely difficult to machine, which is chipboard. Chipboard is much harder material to machine compared to wood (pine, beech, oak) and Medium-density fibreboard (MDF). It is non-uniform, could contain inclusion, resigns, empty spaces or tree bark, all of which cause additional dynamic blade wear. In our studies we compared three different hard coatings (TiN/AlTiN, TiAlSiN and TiN) in order to study, the influence of their structure and composition on their properties, especially in term of durability characteristic on extremely hard to machining chipboard. 

## 2. Experimental Details

### 2.1. Coatings Preparation

The studies described in this paper included TiN single layer coating, TiAlSiN nanocomposite and nanolayer coating and nanolayer—TiN/AlTiN (also named in this article AlTiN). The coatings were deposited on WC-Co cemented blades with 4.5 wt.% cobalt content. The dimensions of the cutting blades were 30 mm × 12 mm × 1.5 mm.

#### 2.1.1. Deposition of TiAlSiN Nanolayer and Nanocomposite Hard Coating [27]

TiAlSiN coating is composed from TiAlN bottom layer, TiSiN layer in the middle and TiAlSiN layer on the top (see optical image of ball crater (callote)—Figure 1 and the cross-sectional TEM image—Figure 2) [27].

All coatings were produced by DC magnetron sputtering in an industrial unit CC800/9 (Cemecon, Würselen, Germany). The deposition system is equipped with four unbalanced planar magnetron sources arranged in the corners of a chamber (Figure 3a). For the preparation of the single layer TiAlN and TiSiN coatings two TiAl and two TiSi targets were active, respectively. The nanostructured TiAlSiN coating was deposited from one pair of TiAl and one pair of TiSi segmental targets (dimensions 88 × 500 mm). The TiAl target was made of titanium with 48 cylindrical aluminum plugs within the race track, while the TiSi target was made of titanium with 17 cylindrical silicon plugs inserted only in one side of titanium target race track. By using double rotation of substrates mounted on the planetary substrate holder system, it is possible to produce nanolayered coatings with uniform thickness of constituting layers. Prior to deposition the substrates were cleaned in detergents and ultrasound, rinsed in deionized water and dried in hot air. Mid-frequency ion etching with bias on turntable of 650 V was conducted for 45 min in mixed argon (flow rate 180 mL/min) and krypton (flow rate 50 mL/min) atmosphere, under the pressure of 0.35 Pa. Additionally, etching was so-called “booster” etching, where the working gas is injected through upper and lower “booster” etch nozzles, where intensive ionization of the working gas occurs. Prior to coating, the deposition chamber was heated to 450 °C. The argon and krypton gas flows were kept constant while the nitrogen gas flow was adjusted during the deposition processes to maintain a constant total operating pressure of 660 mPa. A DC bias of −90 V was applied to the substrates. For deposition of the nanolayered TiAlSiN coating rotation speed of the substrate holder was set to 3 rpm. The detailed procedure of coatings deposition has been presented in the author previous paper [27,28]. TiAlN and TiSiN layers were alternatively deposited to produce a nanolayered and nanocomposite TiAlSiN coating. Thickness of the TiAlN and TiSiN layers was around 5 nm and 2 nm, respectively.

#### 2.1.2. Deposition of TiAlN/TiN Nanolayer Coating

The TiAlN/TiN nanolayer hard coating was also deposited in an industrial unit CC800/9. TiAlN/TiN nanolayer hard coating was deposited from three segmental Ti–Al targets and one Ti target (Figure 3b). Ion etching and deposition parameters were identical to those for preparation of TiAlSiN nanolayer and nanocomposite hard coating. The thickness of individual TiN and TiAlN layers are about few nm and 20 nm, respectively (Figure 4).

#### 2.1.3. Deposition of TiN [28]

Low-voltage electron beam evaporation (or thermionic arc evaporation) system BAI 730 (BAI, Balzers, Vaduz, Liechtenstein was used for deposition of TiN single layer coatings. This system consists of a thermo-emissive cathode (filament), a crucible (evaporation source) connected to a low-voltage supply, and an auxiliary anode (around the target). The plasma (thermionic arc discharge), created between the ionization chamber and auxiliary anode, is used for the heating, etching and evaporation steps. Prior to deposition, the substrates are heated by applying a positive voltage to the substrate table, attracting electrons from the plasma and heating the substrates up to 450 °C. Next, the surface of the substrates is cleaned by ion etching for 15 min. Argon ions are drawn out from the arc discharge and accelerated towards the substrates using a voltage of −200 V. After the etching step, the crucible is made the anode of the arc discharge. The evaporation material held in the water-cooled copper anodic crucible is heated by electrons impinging on the crucible from the thermionic arc. During deposition, the bias voltage on the substrates is −125 V. In order to improve the coating uniformity, the rotating cylindrical substrate holders are placed concentrically around the crucible. Additionally, the crucible moves vertically during deposition to further improve the coating uniformity. The deposition rate of TiN hard coating is about 50 nm/min (for 2-fold rotation of the substrate).

### 2.2. Coatings Characterization

Studies of the coatings surface topography and chemical composition were carried out using scanning electron microscope (SEM) Hitachi SU-70 equipped with a Noran EDS (energy-dispersive X-ray spectroscopy) microanalysis system. The thickness of coatings was estimated from the metallographic cross-sectional SEM images. Scanning transmission electron microscope (STEM) Hitachi HD-2700 and transmission electron microscope (TEM) JEOL JEM 1200 were applied for microstructural characterization of the TiN/AlTiN coating. The cross-section samples for TEM observations were prepared using Hitachi NB5000 focused ion beam microscope (FIB). The TEM microstructure observations were supported by the selected area diffraction (SAED) analysis.

The chemical homogeneity of the produced as received coatings were exanimated by X-ray photoelectron spectroscopy (XPS) method on the PHI 5700/660 Physical Electronics spectrometer. For this purpose, a depth profile procedure was used to the tested coatings by using an PHI ion gun installed in the measuring chamber. For TiN/AlTiN coating, argon ion treatment with an energy of 2 keV was used, while Ar^+^ ion energy of 1 keV was used for TiN and AlTiSiN coatings. An ion energy with value 2 keV proved sufficient to form a crater along the entire Ti/AlTiN coating to the tungsten carbide substrate, while an energy of 1 keV has proved to be insufficient to etch a crater to the WC substrate for the other coatings. The measurements were carried out with the use to of monochromatic Al Kα anode as the X-ray source (hν = 1486.6 eV). During the depth profile procedure, the core lines of O1s, C1s, Al2p, Ti2p, N1s and Si2p were measured. The chemical state of particular elements in coatings was obtained by applying core-line shape profiling method. The averaged calculations of the atomic concentrations were obtained from the analysis of the field under the shape of the core lines by using the MULTIPAK (version 9.8.0.19, 2017, Ulvac-phi Inc., Chigasaki, Japan) software algorithm.

The substrate and coatings’ nano-hardness were determined using the NanoTest Vantage (Micromaterials Ltd., Wrexham, UK) with a diamond Berkovich indenter. The mean hardness was obtained from a minimum of 6 indentations. The loading and unloading time was 20 s. The indentation load of cemented carbide substrate was 1000 mN and 20 mN in the case of coatings. The higher indentation load of the substrate was due to the presence of tungsten carbide grains (typical size of 1–2 µm). It ensured the coverage of a larger area, including many crystal grains with the Co binder interfaces. On the other hand, the lower indentation load is due to layer structure on the top of nanolayer is 80 nm of TiN and 50 nm of AlTiN. In the case of 20 mN indentation load the indentation depth is less than 0.2 µm. Therefore, the indentation response is determined by TiN layer.

The adhesion of the coatings to the substrate was studied using a CSM Instruments RST scratch tester. The indenter load changed from 1 to 100 N over a 5 mm section. The adhesion of the coatings was determined using acoustic emission charts as well as the scratch images. The critical loads (Lc3 defined as the onset of complete coating detachment) were determined. The surface roughness was measured using a Wyko NT9300 light interference optical microscope.

### 2.3. Durability Tests

The durability tests were carried out on standard working center Busellato JET 130. The rotation spindle speed and feed speed amounted 18,000 rpm and 2.7 m/min, respectively, what gives feed per tooth fz = 0.15 mm, and depth of milling 6 mm. The diameter of head produced by FABA company and equipped with one exchangeable blade amounted 40 mm. The dimensions of knives were 29.5 mm × 12 mm × 1.5 mm (N0000814U). Tools are provided from the same producer (FABA). A three-layer standard chipboard with a nominal thickness of 18 mm was used in the blunting procedure. The procedure of detailed durability has been presented in the authors’ previous paper [20]. After each cutting distance of 0.7 m, the milling process was interrupted, and the tool wear was measured on workshop microscope. The tool wear indicator (VB max) was set on 0.2 mm. Information about teet feed and maximum value of wear were estimated on the base of firm Leitz [29]. Figure 5 shows the methodology of measuring wear using the VB max indicator used in this study. This indicator includes both the retraction of the edge and the width of the flank wear.

In the case of a two-edged head, the value of the feed rate given in the Leitz catalog, assuming that the milling depth is about 7 mm, is about 6 m/min. Due to the fact that a single-cutter head was used, this value had to be reduced in the tests to about 3 m/min. This value is in the lower range of the acceptable range due to the very high cutting resistance at the end of the melting period. Due to the vacuum clamping system, the material may shift during processing.

## 3. Results and Discussion

### 3.1. Microstructure, Chemical Composition and Roughness of Coatings

Figure 6 show surface morphology of TiN, TiN/AlTiN and TiAlSiN coatings. Characteristic globules and groove traces originally appearing on the substrate are observed on the surface of all coatings. The largest number and the largest dimensions of globules occur on the TiN coating. On the surface of the TiAlSiN coating, a few defects such as cavities and nodules as well as numerous globules are visible. Table 1 summarizes the surface coating composition data obtained by EDS analysis. The TiN/AlTiN coating contains the largest amounts of aluminum compared to other coatings, which corresponds to the assumed production parameters (Table 1). In the case of the TiAlSiN coating some amount of silicon was introduce (6.2%). The high carbon concentration is due to contamination.

The surface roughness of the TiN/AlTiN and TiAlSiN coatings were similar to each other and to the substrate roughness (R_a_ = 253–295 nm), while the TiN coating had the highest degree of roughness (Table 2) which had a huge impact on the resistance of the blades. The main reason for the significant roughness increase of TiN coating (Figure 7) may be the presence of large nodular structures on the surface as well as the substrate mapping. A sandwich-like coating, in contrast to a uniform coating, does not result in a clear mapping of the substrate, on which the grooves resulting from its surface preparation are clearly visible.

Based on the microstructure images of the metallographic cross-section one can conclude that the TiN and TiN/AlTiN coatings have good adhesion to the substrate (Figure 8). The TiAlSiN coating, due to its high hardness and brittleness, cracked in some places during grinding and polishing processes, which is shown in Figure 8. In the case of TiN coating the Ti/N content ratio is relatively constant on the analyzed surface, while for the TiN/AlTiN coating an abrupt change in the content is observed, which confirms its sandwich structure. The chemical composition on the metallographic cross-section of the TiAlSiN coating is also variable, however, it results from the cracks occurring across the tested track.

Figure 9 shows a cross-section image of a TiN/AlTiN coating. The thickness of TiN/AlTiN coating is approximately 4.5 µm. It has a columnar microstructure. The individual column with typical diameter around 200 nm, is composed from of several grains that are elongated in the growth direction. The microstructure of the coating near the substrate consists of finer grains. Their growth is repeatedly interrupted by a re-nucleation process. The SAED (selected area electron diffraction) pattern proves that it has a fcc crystal structure (see Figure 9). The layer 50 nm thick semi-transparent AlTiN and the bottom 80 nm thick reflective TiN layer were added on the top of nanolayer structure in order to provide a blue color of the coating.

### 3.2. Chemical Composition of the Tested Coatings

In Figure 10 the depth profile results collected for all study coatings have been shown. The result recorded for TiN/AlTiN coating consists of three clearly separated zones (Figure 10a). The carbon and oxygen surface contaminations of studied coating were totally removed after about 10 min of ion treatment. In the first zone in the range of 10–50 min of ionic etching, the content of Al is over 30%, the Ti content varies at 15% and the N content is about 50%. Therefore, the zone can be marked with the symbol AlTiN to show that the content of Al is greater than the Ti content. The atomic concentration seems stable on the same level for all components. In the so-called transition zone located in range of 50–100 min a sharp decrease of Al content and parallel sharp increase of Ti can be observed. The amount of nitrogen remains unchanged due to the lack of aluminum target sputtering in the initial phase of coating growth. The average content of Ti and N in the range of 100–120 *x*-axis scale is about 45% for both components. It is worth noting the very low oxygen and carbon contents of both coatings at 3%–4% level. The average atomic concentration calculated for zones 1 and 2 were shown in the Table 3. After about 130 min of argon treatment, the third transition zone begins in which the Ti and N content sharp decreases while W and C components related to the WC substrate rapidly increases.

Figure 10b shows the depth profile obtained for components of the TiN coating. The first zone consists small amounts of carbon and oxygen contamination on the about 5% level, however second zone indicates on the stable concentration of the main components of TiN coating with concentration of 51% for nitrogen and 47% for titanium in the cross-section of the layer. The contamination level of about 1%. The results of the averaged atomic concentration the entire shell section for zone two were presented in the Table 3.

The depth profile results obtained for last studied coating AlTiSiN was presented in Figure 10c,d. In the first zone, up to about of 120 min of argon etching, a decrease of carbon and oxygen contaminations were observed with a simultaneous increase of the components of the analyzed coating. In the second zone, the concentration of Al, Ti, Si and N is 8%, 35%, 3% and 50%, respectively. We concluded that in this part of the analyzed coating the AlTiSiN layer existed. The third zone is a transition area, for which a gentle increase of titanium and silicon contents and a decrease of aluminum and carbon concentrations were observed. A fourth zone containing 51% nitrogen, 42% titanium and 5% silicon constitutes the nanocomposite structure of the TiSiN layer. It is worthwhile to mention the very low level of oxygen and carbon contamination of actual coating at 2–3%, as was indicated in the Table 3.

Figure 11 shows the XPS spectra of the main components of the analyzed coatings. The N1s core lines consists of two components for all of the tested samples. The main component with higher intensity can be ascribed to nitride while the second to shake-up photoemission. In case of the AlTiN outer part of coating (see Figure 11a) the position of the main component is shifted towers to lower binding energy by 0.2 eV. The position of the shake-up peaks is slightly different for all the tested coating. Basically, the Ti2p doublet contains three components that can be attributed to the presence of TiO_2_, TiON, and AlTiN or TiN in the tested coatings [28]. The position of the Al2p line is the same for AlTiN and AlTiSiN coating. In the case of the AlTiN coating, the Al2p line shape includes a component derived from Al_2_O_3_ oxide [28]. The presence of this oxide cannot be ruled out in the AlTiSiN coating because it is indicated by the small amount of oxygen of about 1% given in Table 3. Nevertheless, its amount may be marginal. The shape of the Si2p line was the same for presented in Figure 10 both zone II of the AlTiSiN coating and zone IV corresponding to the part of the TiSiN. The line shape structure contains two components that can be attributed to the presence of Si atoms in stoichiometric Si_3_N_4_ and non-stoichiometric SiN_x_Si nanoshells [30]. In our opinion an atomic concentration at zero for oxygen component in the Table 3 excludes the presence of TiO_2_ nanocomposites in the TiSiN coating structure.

### 3.3. Mechanical Properties and Durability Tests

The hardness of a blade is as important as its wear resistance. The results of nano-hardness of the sintered WC/Co substrate as well as the nano-hardness of TiN, TiN/AlTiN and TiAlSiN coatings are presented on the Figure 12. It could be observed that the nano-hardness of the single layer TiN coating demonstrated the highest nano-hardness (29.34 GPa) among all investigated materials, whereas the substrate had the lowest nano-hardness (24.41 GPa). However, it should be emphasized that very high values of the standard statistical error mean that the values, despite the differences, are similar and may indicate a certain heterogeneity of the material. These findings indicated that all of the coatings could enhance the substrate nano-hardness. Analysis of the obtained results showed that the TiN/AlTiN coating with a high aluminum content is characterized by the lowest nano-hardness among the analyzed coatings, comparable to the substrate and with the smallest statistical standard error. Compared to the results of nano-hardness available in the literature, the slightly discrepancies are observed. For example, TiN hardness on WC-Co is ~18.5 GPa [6], and our TiN deposited also on WC-Co is 29.4 GPa. This divergence may be due to the applied indentation load. In our case, we deliberately applied different types of indentation load (different for coatings, different for the substrate) to test the effect of individual layers on hardness.

The nanoindentation results of TiN/AlTiN coating are not in agreement with durability tests (Table 4) where this coating had the best characteristics. This can be due to the multilayered structure of this coating which differ the mechanism of wear in this kind of material. Analysis of durability test results shows significant differences in performance test of the coatings. This is reflected in cutting distance that was 7083 m for TiN/AlTiN coating. This value is much higher than in case of reference uncoated blade (4909 m). Inversely, in the case of TiN and TiAlSiN coatings, despite a little higher nano-hardness of TiAlSiN, the TiN layer is characterized by better wear properties (5825 m) in comparison with TiAlSiN layer (5694 m). This phenomenon confirms common opinion that in case of materials used in wood industry the increase of nano-hardness does not mean increase of tool durability. The crucial factor that has got dominant role is fracture toughness. The reason of this relationships is specific mechanism of chipboards cutting. Especially, mineral contaminations cause very high dynamic loading of edge what in consequence lead to chipping and damages of cutting edge. Taking into account the fact that spread of results in case of TiN/AlTiN is only slightly higher than for substrate, this kind of coating can be regarded as very promising.

The critical loads (Lc_3_ defined as the onset of complete coating detachment) are collected in the Table 5. The TiAlSiN coating is characterized by high critical load Lc_3_ of 32.8, respectively, indicating their applicability in the tool industry. In contrast, in the case of TiN coating the complete coating separation from the substrate is recorded at low loads Lc_3_ = 13.4 N which indicates poor layer adhesion to the substrate and could be cause of the insufficient characteristics of these coating in durability tests (Table 5). Coating adhesion tested at the diamond indenter -coating interface does not properly reflect the behavior of the blade in contact with the chipboard. The destruction mechanism during scratch-test is different than when working with chipboard.

## 4. Conclusions

The multilayer TiN/AlTiN coating have shown the best durability characteristics tested on extremely difficult to process chipboard (the highest cutting distance: 7083 m in comparison with TiN: 5825 m and TiAlSiN: 5694 m).

This is strongly related with their microstructure, surface roughness and chemical composition. The specific structure of the coating, namely the presence of sublayers TiN/AlTiN turned out effective in obtainment of the high durability and simultaneously maintaining of not very high spread of cutting distance.

The coatings nano-hardness is not the crucial factor for tools durability. The dominant role is fracture toughness which is related with the internal structure of the TiN/AlTiN coating.

The opposite results were observed in the case of nano-hardness of TiN and TiAlSiN layers. The higher nano-hardness of TiAlSiN have not caused higher durability. 

Scratch tests in case of TiN layer turned out completely unsuccessful. This may be due to the different. Coating adhesion tested at the diamond indenter-coating interface does not properly reflect the behavior of the blade in contact with the chipboard. The destruction mechanism during scratch-test is different than when working with chipboard.

The disadvantageous test results in the case of TiN layer can be cause by the highest surface roughness that could have had the influence on durability and scratch tests.

The most important parameter of coatings from the point of view of their applicability is durability. Among the three analyzed coatings, the TiN/AlTiN coating was characterized by the best durability. All three coatings had better durability than the substrate.

Another direction of research is to modify the substrate by blasting before and after coating.

## Figures and Tables

**Figure 1 materials-15-07159-f001:**
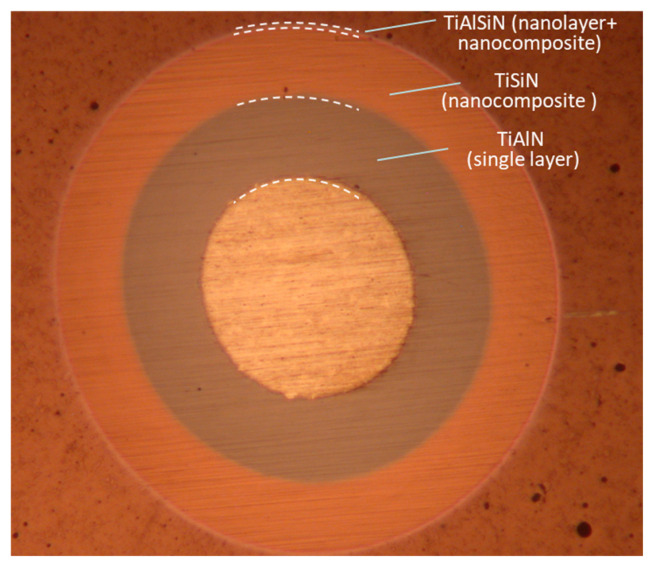
Optical image of ball crater (calotte) through entire »TiAlSiN« hard coating [27].

**Figure 2 materials-15-07159-f002:**
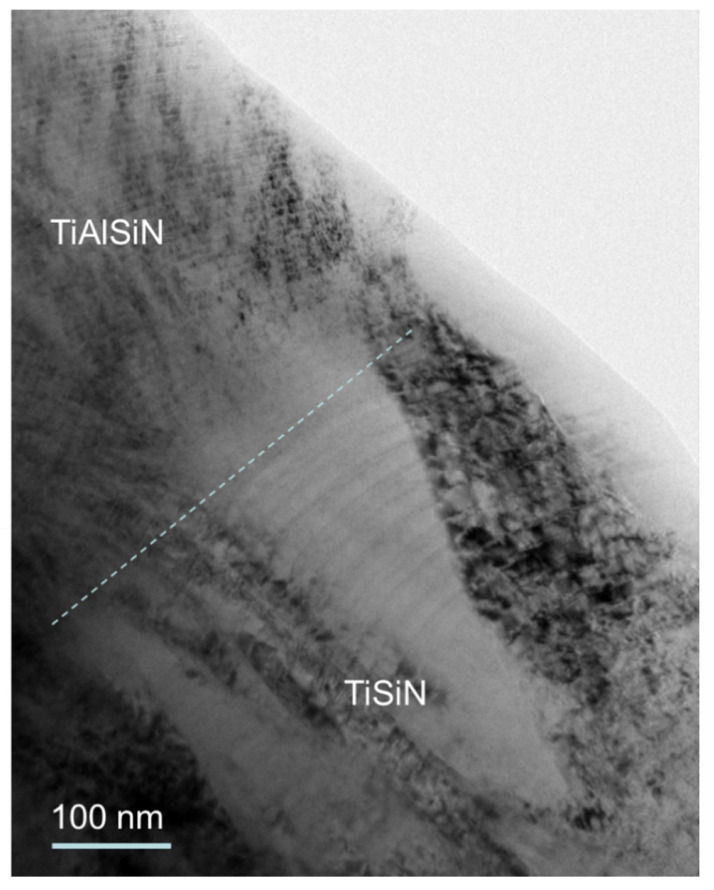
Cross-sectional TEM image of nanocomposite TiSiN and nanolayer TiAlSiN layers in »TiAlSiN« hard coating [27].

**Figure 3 materials-15-07159-f003:**
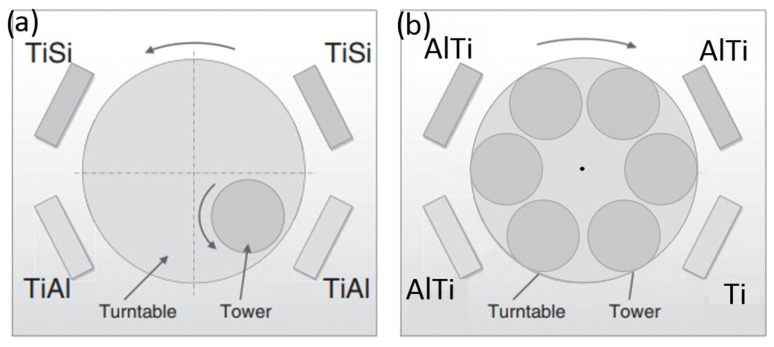
Configurations of the targets for the deposition of the nanolayer: (**a**) TiAlSiN and (**b**) TiN/AlTiN hard coatings.

**Figure 4 materials-15-07159-f004:**
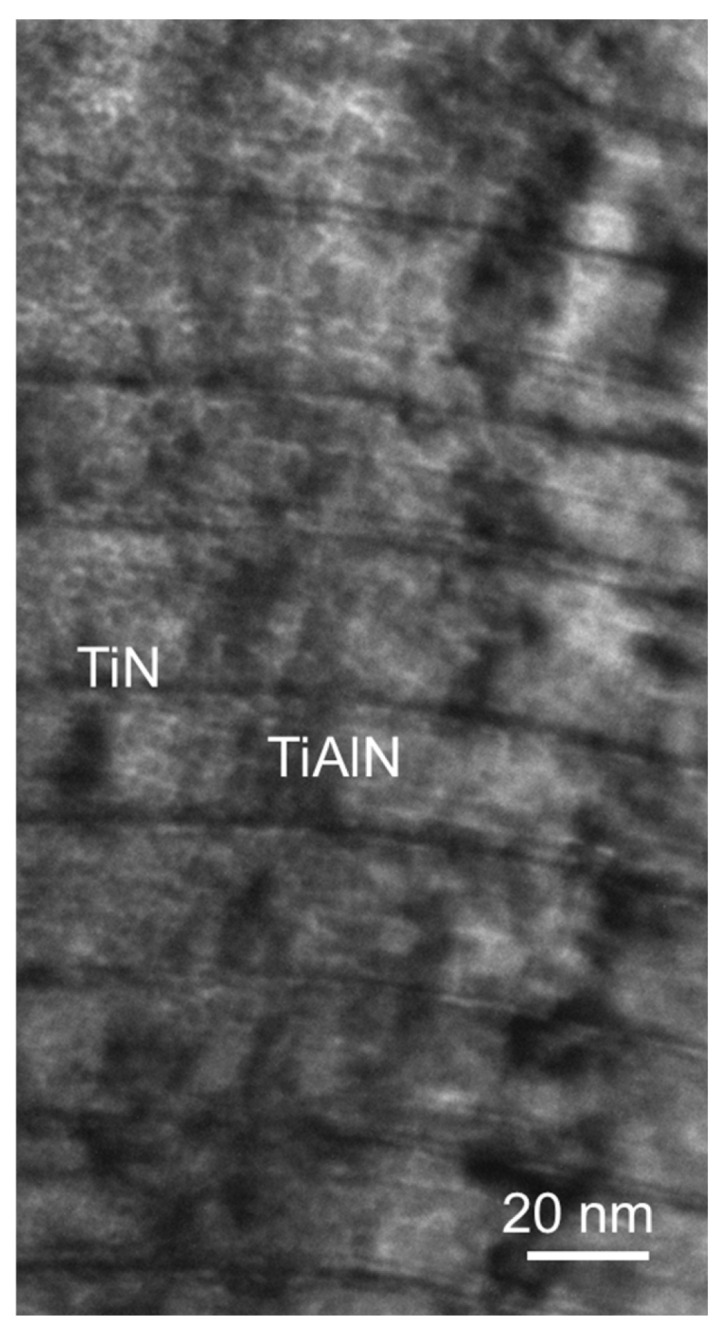
Cross-sectional TEM image of TiAlN/TiN coating.

**Figure 5 materials-15-07159-f005:**
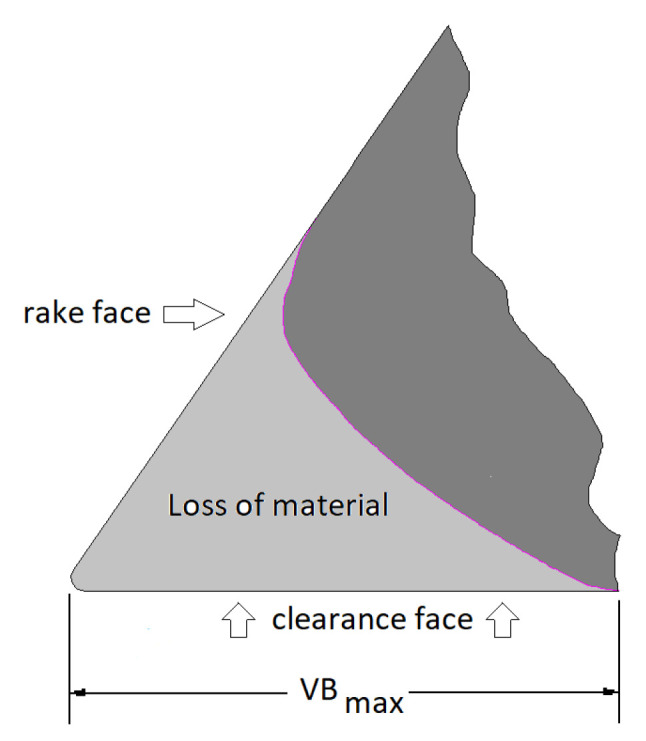
Schematic illustration VB_max_.

**Figure 6 materials-15-07159-f006:**
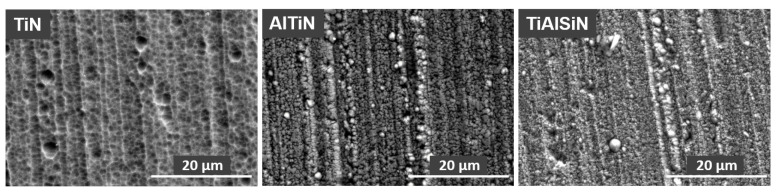
Surface morphology of TiN, AlTiN and TiAlSiN coatings.

**Figure 7 materials-15-07159-f007:**
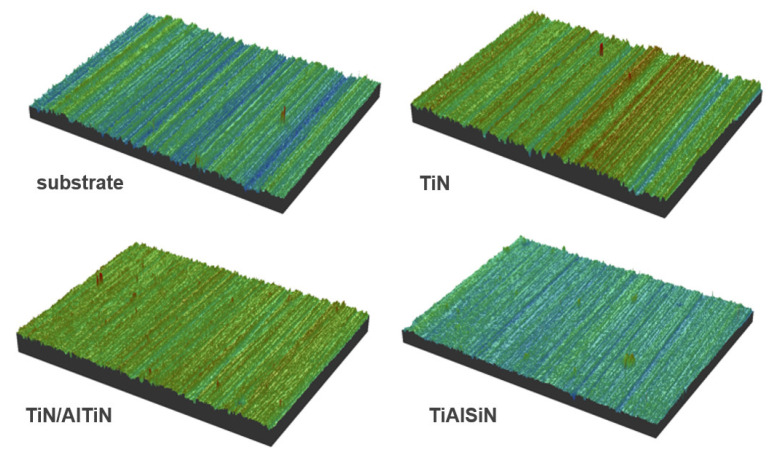
Three-dimensional surface topographies of bare cemented carbide substrate and the same substrate after deposition of TiN, TiN/AlTiN and TiAlSiN coatings.

**Figure 8 materials-15-07159-f008:**
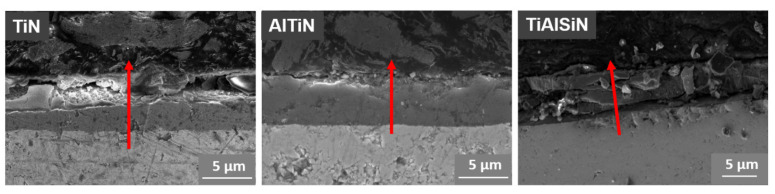
Cross-sectional SEM images of TiN, TiN/AlTiN and TiAlSiN coatings.

**Figure 9 materials-15-07159-f009:**
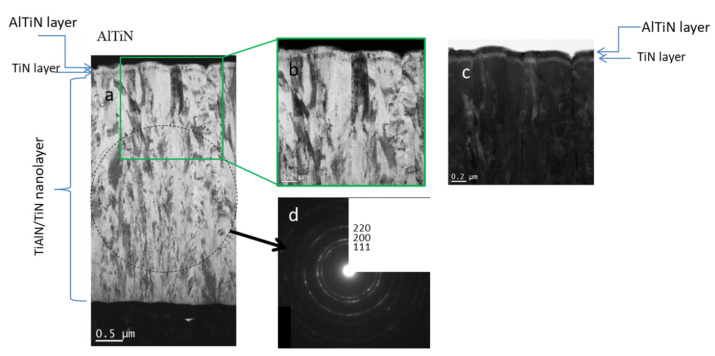
The cross-sectional BF-STEM image at lower (**a**) and at higher magnification (**b**). STEM image in Z-contrast mode (**c**) and the corresponding electron diffraction (**d**) of TiN/AlTiN coating.

**Figure 10 materials-15-07159-f010:**
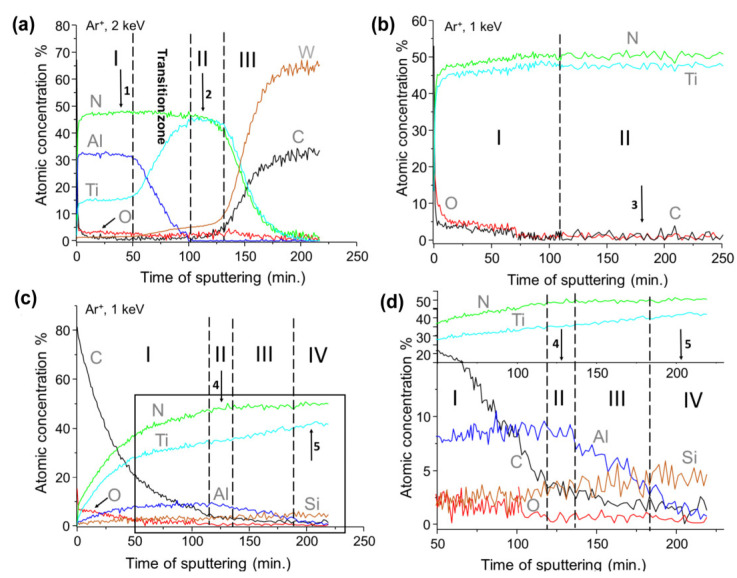
The depth profile analysis of the TiN/AlTiN (**a**), TiN (**b**) and TiAlSiN (**c**) coatings. The enlarged region of the depth profile of TiAlSiN was shown in figure (**d**). The arrows indicate points in the depth profile where the core lines were chosen to analysis. Areas of the ion treatment marked with Roman numerals are described in the manuscript.

**Figure 11 materials-15-07159-f011:**
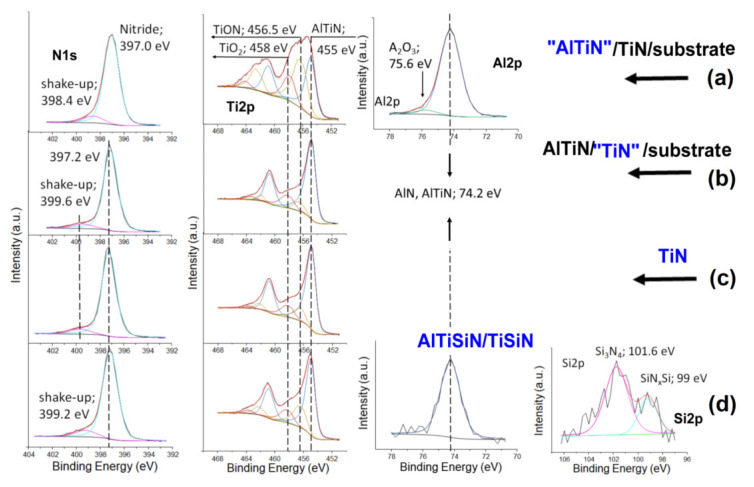
The N1s, Ti2p, Al2p and Si2p core lines recorded from the tested coatings—(**a**) “AlTiN”/TiN, (**b**) AlTiN/”TiN”, (**c**) TiN and (**d**) AlTiSiN/TiSiN. The presented spectra were chosen from the inner region of the tested coatings marked by an arrow in Figure 10. The blue color indicates the type of shell for which the core lines are presented.

**Figure 12 materials-15-07159-f012:**
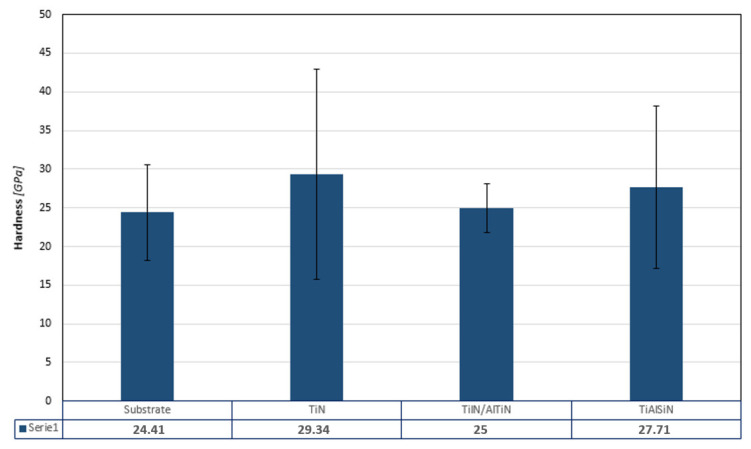
Nano-hardness of substrate and TiN, TiN/AlTiN and TiAlSiN coatings.

**Table 1 materials-15-07159-t001:** Chemical composition of TiN, TiN/AlTiN and TiAlSiN coatings by mean of an EDS analysis.

Material	Atomic Concentration % of the Particular Elements of the Studies Coatings							
	W	Co	Ti	Al	N	O	C	Si
Substrate	54.1	2.7	—	—	—	6	37.2	—
TiN	—	—	61.3	—	38.7	—	—	—
TiN/AlTiN	—	—	25.3	26.1	34.2	2.3	12.2	—
TiAlSiN	—	—	41.1	5.1	35.4	—	12.2	6.2

**Table 2 materials-15-07159-t002:** Surface roughness parameters of substrate, TiN, TiN/AlTiN and TiAlSiN coatings.

Material	Ra [nm]	Rq [nm]
Substrate	253	317
TiN	414	532
TiN/AlTiN	295	377
TiAlSiN	279	373

**Table 3 materials-15-07159-t003:** Chemical composition of the tested coatings obtained from collection analysis of the XPS spectra. The averaged results of atomic concentrations calculations given in the table are related to the bolded part of the coatings description.

Type of Coating	Atomic Concentration % of the Particular Elements of the Studies Coatings					
	C1s	N1s	O1s	Al2p	Ti2p	Si2p
**AlTiN**/TiN	1	47	3	33	16	-
AlTiN/**TiN**	4	45	4	-	47	-
TiN	1	51	1		47	
**TiAlSiN**/TiSiN	3	50	1	8	35	3
TiAlSiN/**TiSiN**	2	51	-	-	42	5

**Table 4 materials-15-07159-t004:** Durability test results of substrate, TiN/AlTiN, TiN and TiAlSiN coatings.

Material	Cutting Distance [m]	SD
Substrate	4909	537
TiN	5825	673
TiN/AlTiN	7083	927
TiAlSiN	5694	754

**Table 5 materials-15-07159-t005:** Critical load values for TiN, AlTiN and TiAlSiN coatings.

Material	Lc_3_ [N]
TiN	13.4
TiN/AlTiN	14.1
TiAlSiN	32.8

## Data Availability

Not applicable.

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
