# Peer review of "Comparison Study of PVD Coatings: TiN/AlTiN, TiN and TiAlSiN Used in Wood Machining"

_materials, 2022, doi:10.3390/ma15207159_

Round 1

Reviewer 1 Report

Increasing wear resistance of cutting tools is an important practical task aiming the extension the potential of using wood industry machines. Promising solutions include special tool surface treatment (e.g. ion nitriding) or hard protective coatings (physical or chemical vapor deposition).

The current work of B. Kucharska et al. was carried out within the framework of the latter approach and is devoted to the study of hard coatings deposited onto cutting blades made of sintered WC-Co. Three compositionally different coatings (TiN, TiN/AlTiN, TiAlSiN) of approximately the same thickness were obtained by slightly different physical vapor deposition procedures. The paper presents the data on the microstructure (SEM, TEM), surface roughness (optical microscopy) and composition (EDS and XPS) of these coatings, as well as some functional characteristics (hardness, scratch and durability tests). However, the results are described and presented insufficiently, which does not allow a clear definition of their reproducibility, novelty and scientific significance.

The following major amendments should be made to understand the current work:

1) The Introduction and references in it mainly cover the research area of hard coatings based on chromium nitride (lines 46-58). However, the authors investigated the coatings based on titanium nitride. Moreover, the introduction ends with information about the addition of chromium nitride to TiAlN coatings (lines 73-75), which is not reflected in any way in the further text of the article.

Thus, the Introduction should be seriously rewritten with a clear focus on the objects of current research. The scientific landscape in the field of titanium nitride hard coatings should be clearly presented. It should reflect the current trends in multi-layer coatings on this basis and in the preparation of coatings of mixed composition (addition of aluminum, silicon, etc.). The final sentences of the Introduction must clearly indicate the scientific novelty of this study.

2) Likewise, a comparison of the results obtained with the relevant literature should be added in the Discussion. It is now completely unclear what discoveries this study has brought to this rapidly developing field.

3) It should be clearly (point-by-point) indicated the differences of TiN/AlTiN coatings obtained in this work from those obtained and studied in the same way in the previous work of these authors (https://doi.org/10.3390/ma14112740). Note that this work is not included in the bibliography. This once again brings us back to the question of scientific novelty.

4) The preparation procedures for all three types of coatings must be clearly described in accordance with the requirements of reproducibility. Now section 2.1 does not even give an idea about (a) how the TiNi coating was prepared (b) what is the fundamental difference in the approach to obtaining "nanocomposite" (TiAlSiN) and "nano/multilayer" (TiN/AlTiN) coatings.

5) The terms used are confusing. In particular, why is the TiN/AlTiN coating called "nanolayer" or often even "multilayer"? According to TEM data, there is no pronounced layering in its cross-section microstructure, with the exception of layers specially added on top (Fig. 5). According to XPS data (Fig. 6), there are only two extended layers, the thickness of which cannot be attributed to the "nano" category.

Why is TiN called a "single-layer coating", and AlTiSiN called a "nanocomposite coating". Table 3 shows the two layers of the last coating (AlTiSiN). How is this fundamentally different from the TiN/AlTiN coating (also two layers according to Figure 6)?

Thus, it is necessary to verify and unify the terms.

6) XPS data are presented sloppily. What was the calibration? At such points/intervals, the quantitative compositions of the coatings are given in Tables 3 and 4?

In addition, it is necessary to add information about the chemical forms of the elements at key points along the depth of coatings (results of core level fitting). The fitting curves can be added to the Supplementary Materials, but the chemistry should be discussed in the text of the manuscript.

7) The authors claim that the high carbon content (Table 1) is due to pollution (line 164). It should be clarified what kind of pollution is meant. Probably, not only the surface, since a noticeable amount of carbon also appears in the composition of the AlTiSiN coating after etching (Table 3). The origin of this carbon should be indicated.

In addition, the purity of the original targets and the manufacturer of the substrates should be indicated.

8) It is necessary to add the phase composition of all coatings (XRD data), and not just the chemical one.

9) It should be more clearly related to the roughness of the coatings with their microstructure, and, accordingly, with the methods of obtaining. In particular, the explanation is desirable, why given a comparable thickness, "sandwich-like coating, in contrast to a uniform coating, does not result in a clear mapping of the substrate" (lines 178-179)? What do the TEM data show in this aspect? Does re-nucleation take place precisely in the region where the composition of the TiN/AlTiN coating changes? Were there no "repeatedly interrupted re-nucleation" in the process of obtaining TiN coating? Then it is not the layered structure that is the cause of the observed effect, but the technological features of the preparation.

10) Why did the authors obtain the coatings with a thickness of about 4.5 µm? What thicknesses are optimal for the described application? Add relevant literature.

11) The results of Figure 13 do not show a significant difference in the hardness of the substrate and coated samples. As this is an experimental fact, the description should be adjusted accordingly (lines 258-266).

12) The authors conclude that TilN/AlTiN coatings is superior in terms of durability test results (Table 5). However, the critical load value for this coating is almost the same as for TiN, which exhibits poor adhesion (Table 6, line 291). This value is much lower than that of TiAlSiN coating is characterized by "high critical load … indicating their applicability in the tool industry" (lines 288-289).

Therefore, the best coating of this article is not applicable to the industry. This should be clarified and directions for further work should be indicated in order to obtain applicable coatings with better performances.

13) The Conclusions should be reformulated, in accordance with the above comments.

14) Correction of English (singular/plural) and technical typos in the text is required.

Some such errors clearly follow from a previous article by the authors: for example, Table 5 "TiAlN/a-C:N coatings deposited on different substrates" (in the current article there are no coatings of this composition and the substrates do not vary).

It is necessary to correct the names of sections ("3.2. Chemical and structural composition of AlTiN coating" - no, there are for all coatings), Tables and Figures (indications for specific samples are often lost). Remove duplicate phrases (for example, lines 211-214).

Reviewer 2 Report

The comments are as follows:

1. The review of previous research can be supplemented with new references. I think 14 references is not enough for scientific research. Additionally, the manuscript cited a reference [15] that does not exist in the literature. Reference [14] not cited.

2. The manuscript is missing the last paragraph in the Introduction section. In that paragraph, the shortcomings of the previous researches should be stated and based on that, the following should be defined: the goal of the research, the innovation of the methodology, the scientific contribution and the scientific hypotheses.

3. In the complete chapter "2 Experimental details" various parameters are listed, but their choice is not justified. Why and on what basis they were chosen. It would also be desirable to accompany this chapter with some additional photo images.

4. Why is the tool wear indicator (VBmax) set to 0.2 mm?

5. In the discussion, can the obtained results be compared with previous similar researches?

6. A cost analysis would be desirable.

Round 2

Reviewer 1 Report

The authors significantly revised the manuscript and added new clarifying data, following the recommendations of the reviewers. In general, the updated version of the article is a rather interesting qualitatively performed study, the main drawback of which is the lack of XRD data. However, the elemental composition of the obtained coatings has been satisfactorily proven.

At the same time, there are still a few major comments that should be corrected before publication.

1) The authors added that the "chemical stability" of the coatings was investigated using XPS (line 227). It should be clear what is meant by this. Was the special maximum strong mode of the ion gun applied? If so, how does this relate to the postulated application of coatings? Perhaps a study of coatings was carried out before and after mechanical tests? However, this is not reflected in the manuscript. Finally, it is likely that the authors investigated not "chemical stability" but "chemical composition by depth of coatings."

2) The added information shows only the figures of the fitting of the individual XPS peaks of the elements (Fig. 11). However, there is no description. This is unacceptable. Authors should indicate what the major and minor components of the fitting correspond to, and their relationship. If the authors consider that this detracts from the main idea of the article, then they can put this information in the Appendix or Supplementary Materials.

In addition, fitting information is available only for one type of the studied coatings. This should be added for other coatings, as it is a clear indication of the chemical forms of elements in the absence of XRD data.

3) The size, thickness and other important characteristics of the cemented carbide substrates used should be clearly stated.

4) The statement remains that "TiN/AlTiN coating with a high aluminum content is characterized by the lowest nanohardness among the analyzed coatings, comparable to the substrate" (pieces 410-412). However, the data in Figure 12 clearly show that all differences between the obtained nanohardness values cannot be considered reliable from the point of view of standard statistical processing. Thus, the specified statement should be replaced by a milder version.

5) The title of the article postulates a "comparison" of three types of coatings. Accordingly, the results of this comparison should be clearly reflected in the Conclusions. Indicate which coatings were the best and by what parameters. Please specify directions for further improvement of coatings in accordance with the stated application.

Reviewer 2 Report

The manuscript has been updated.

Author Response

Dear Reviewer,

Thank you one more time for reading our manuscript and reviewing it.

Best regards,

Beata Kucharska